# Profile-Guided Quantization: A Compiler Solution to Automate Quantization for Efficient LLM Training

Gil Tabak, Clemens JS Schaefer, Xiaofan Zhang, Denali Molitor,
Jinliang Wei, Zongwei Zhou, Philip G Hendrix, Mitchelle Rasquinha
Google LLC

*Abstract*—The growing size and complexity of Large Language Models (LLMs) for generative artificial intelligence (AI) have significantly intensified the compute and memory demands during training and serving. While scaling up model size has fueled rapid progress to deliver advanced AI capabilities, it is increasingly challenging to efficiently host these models on hardware where resources are always constrained. As a promising compression technique, quantization is widely used to address hardware efficiency bottlenecks. However, how and where to apply quantization remains challenging. Significant barriers complicate the practical use of quantization ranging from varied developer skill in applying quantization, to diverse numeric sensitivity of targeted AI models, and different quantization support from ML frameworks and back-end hardware. To address these issues, we propose profile-guided quantization (PGQ), a compiler-based solution that leverages logged tensor statistics and metadata to automatically determine optimal quantization settings for individual operations given targeted workload and hardware. PGQ alleviates the domain knowledge needed from model, framework, and hardware and streamlines the design and implementation of quantization recipes through graph-level instruction rewriting within the compiler, which makes quantized LLM training accessible to a much broader audience. As an example of our proposed approach, we demonstrate its application in quantized training use cases with GemmaV2 models showing up to 18.2% speed up of a training step.

## I. INTRODUCTION

Recent developments in Large Language Models (LLMs) showcase their significant advancement in generative artificial intelligence (AI) capabilities [6], [27] along with the drastically increased compute and memory demands [26]. Therefore, techniques to deliver model compression and efficient hardware utilization become the key to keep AI development progressing. Quantization is one of the promising techniques to achieve this goal by reducing the numerical precision of model parameters and its arithmetic operations. This is especially relevant as modern hardware accelerators increasingly support low-bit formats [10], [16] and ongoing research explore ultra low-bit designs to maximize hardware efficiency while maintaining quality [7], [17].

In large-scale industry settings where models are rapidly developed and updated, adapting quantization techniques remains challenging, despite promising research results across various bit-widths (from ultra-low [7], [17] to 8-bit formats [4], [18], [24], [31], [33], [36]) and quantization settings (post-training, quantization-aware, or fully quantized training). Challenges include the considerable amounts of user knowledge required to perform framework level model changes to orchestrate the optimal use of quantization techniques. This entails model-specific knowledge and individual hyperparameter tuning where each change requires new evaluation of the performance-quality trade-off.

Further, there is often diversity in ML frameworks [5], [19], [29] with varying levels of API support for quantization. Quantization experts often use solutions implemented at the framework level, which results in additional difficulties for maintenance, since frameworks also tend to evolve rapidly. Additionally, once suitable quantization configurations are found they tend to short-lived due to newer models being deployed, as well as new hardware capabilities.

The complexity in pinpointing optimal quantization choices is further compounded by the rapid and expansive research progression of quantization techniques. Recent advancements have introduced a plethora of improved methods for different parts of the quantization process, e.g. rounding schemes [9], clipping methods [21], or novel quantization formats [2]; however, it remains unclear how to determine which quantization methods to apply (and potentially combine), or even which specific operations to quantize. We refer to these choices as the quantization *recipe*. Recipes for model compression may apply to weights only, recipes for serving use cases may quantize both weights and activations, and finally recipes for training may quantize weights, activations, errors, and gradients.

The selection of the appropriate bit-width or potentially use of additional techniques presents search problem with a exponentially growing search space with the number of operations to be quantized. Even with simplifying assumptions like independence of impact, assessing the impact of individual operations may be prohibitive. For example the authors of [33] proposed a layer-wise quantization approach (assuming independence of impact) to tackle the challenge of an exponentially large search space in mixed precision model optimization. However, their method remains computationally intensive for practical applications since it requires testing every layer individually. Alternative approaches have leveraged sensitivity metrics and trade-offs with performance gains to identify Pareto-optimal models, often incorporating second-

Fig. 1: **Illustration of the three-step process.** A visual guide for the PGQ workflow discussed in Section II. Blue shows steps, yellow represents data/recipes, and green shows the user-specified policy.

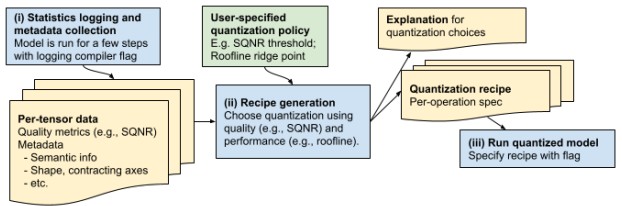

order information like Hessians [12], [13], [35]. Inter-layer dependencies further refining second-order sensitivity metrics have been considered in [23]. However, such approaches are often computationally expensive, require deep integration with the model code, and are not applicable for all tensors in training, such as gradients. In parallel, quantization-aware training techniques have been explored to learn optimal bit widths directly during model training [22], [30]. Noticeably, many of the methods mentioned only apply to a subset of all operations required during training, and bear additional computational costs. Additionally, they require tight integration with the model code, making them more difficult to apply broadly.

In this work we propose a general profile-guided quantization approach which improves over existing quantization approaches in three ways: (i) We propose the use of 'local', easy to compute metrics to assess the difficulty of quantization (Section II-B), and focus on signal-to-quantization-noise (SQNR). A single local metric narrows the search space substantially and offers the ability to reason about quantization choices in a white-box fashion. (ii) We define a methodology to assess the quality of quantization metrics (Section II-D). (iii) We introduce a comprehensive quantization implementation via the compiler, including a stage for logging operations to generate local metrics, as well as applying the actual quantization. This approach is designed to be user-friendly as it requires no changes to the model (such as Gemma, Llama, or DeepSeek) or framework code.

## II. PGQ METHODOLOGY

PGQ is separated into three stages shown in figure 1: (i) statistics logging and metadata collection, (ii) recipe generation, and (iii) quantized model rewrite. In our implementation, stages (i) and (iii) are integrated directly into the compiler, thereby not requiring changes to the model or framework level code. Stage (ii), recipe generation, depends on metadata which is generated in stage (i). Metadata includes semantic type of operation and inputs (e.g., weight, activation, or gradient), sharding information (in the case of model parallelism), and other information (e.g., which dimensions are contracting in a general dot operation). The recipe generation (stage ii) also produces an explanation of which operations were quantized, in which way, and why the choice has been made to foster user trust.

PGQ can take into account performance using the roofline model [32] when creating recipes in stage (ii) using the metadata information collected in stage (i) and the ridge point of the accelerator. PGQ also takes into account 'local' quality metrics to assess the impact on quantization. Specifically, we focus on the signal-to-quantization-noise ratio (SQNR) showing its usefulness in Section II-B. We found relatively little logging data (a few steps) is sufficient to obtain meaningful SQNR values.

PGQ is not a specific quantization method, but rather can be integrated with various approaches to improve quality (see Section II-F). Different formats, subchannel sizes, and quantization techniques will be reflected in the local quality metrics.

### A. Scope of quantization in this paper

The types of operations we focused on are general matrix multiplications (GEMM) including dot general and convolution operations, for which recent specialized accelerators have been developed. Throughout the paper, we focus only on bf16 (unquantized) and int8 (quantized) formats . The scaling factors used for quantization are determined individually for separate non-contracting dimensions. Their values are found using symmetric quantization without clipping, i.e., the component with largest magnitude (absolute maximum value) within each axis is mapped to the largest representable value in the target format.

### B. Local Metrics and SQNR

In contrast to the computational complexity entailed with network wide (global) metrics, local metrics rely on data distribution within tensors and come with their own set of challenges. The primary challenge is that the distribution of data across tensors may vary widely. This variance can be across different types of networks, different components within networks, differences in the inputs, and differences between the types of nonlinear layers and normalization techniques used (for example see Figure 3 below).

Possible choices for local metrics range from simple statistics such as variance to sophisticated information-theoretic measures. We use the signal-to-quantization-noise (SQNR) as our primary local metric. We validated SQNR empirically and found that it is informative (Section II-D). The SQNR is defined as:

$$\text{SQNR}(X) = \frac{X^2}{\mathbb{E}\left[\left(\mathbb{Q}(X) - X\right)^2\right]}. \tag{1}$$

Where $\mathbb{Q}$ is the quantization operation and $X$ the underlying tensor. SQNR is closely related to the mean-squared error (MSE), which is the denominator here. However, the SQNR is scale-invariant, which makes it more suitable for comparing different tensors in a neural network thereby enabling a network wide quantizability ranking of tensors. This is because the usable information content of given tensors should not depend on the absolute scale of the tensor values, given the scaling mechanism used in quantization.

Furthermore the SQNR of a given tensor may change over time when training a model. This change is usually most detectable during the start of the pre-training phase, and less detectable during fine-tuning. We generally recommend to monitor SQNR, or only apply quantization after training has somewhat stabilized.

### C. Using Metrics for Recipe Generation

To generate a quantization recipe based upon a local metric we consider the case where each operation is either quantized at some precision, or not quantized. The main idea is to simply order the operations by a local metric like SQNR, and pick a threshold at which an operation will be quantized. SQNR can be used at both the inputs or the outputs of operations. For simplicity, we use the SQNR of the inputs in order to avoid the extra compute and complexity.

Notice that importantly the threshold is not known a priori. It may vary significantly among models, use cases, etc. In practice many use cases tend to have similar thresholds, and usually only a few trials may be needed to find a suitable threshold.

### D. Adversarial Recipes: Towards Empirical Evaluation of Metrics

We propose to use local metrics for *ordering* of the operations, not for absolute quality statements. In general, it holds that when more layers are quantized performance will improve and quality will degrade (see Figure 2). In order to assess the quality of a given metric we are interested in approximating the Pareto curve as closely as possible.

In Figure 2 the Pareto curve represents the 'best' hypothetical ordering possible, whereas a good metric will produce a similar curve. In practice, it is not often possible to produce the Pareto curve with certainty, because the search space grows exponentially with the number of operations. To further validate any metric, the order given by the metric can be reversed to produce an 'adversarial' curve (shown in Figure 2). We can make assessments about the quality of a metric by comparing the original versus adversarial curves. A significant gap between the two curves suggests that a given metric is informative.

### E. Stochastic Rounding

Application of stochastic rounding is known to be helpful when using low precision formats for some operations, particularly for the gradient inputs [9], [14], [25].

In general, gradients tend to have lower SQNR when using formats of limited dynamic range (such as low bit width integer formats), because of their heavier-tailed distributions (see Figure 3 for examples). However, the application of stochastic rounding can make training more robust to lower SQNR for gradients. For this reason, when using SQNR to decide which operations to quantize, we generally use separate criteria/thresholds for forwards versus backwards operations with stochastic rounding.

Fig. 2: **Illustration of the impact of various orderings on quality and performance.** The curves represent different orderings of operations being quantized (for simplicity we assume there is a binary choice between quantizing or not). The upper left corner represents no quantization, where quality is highest and speed is lowest. The bottom right represents the case when all operations considered are quantized. Each curve begins and ends at the same two points, however the quality and step time trade-off may be vastly different. The top curve illustrates the 'optimal' curve, in the sense that the ordering it represents corresponds to the Pareto optimal trade-off between quality and step time. The other curves illustrate a 'good' ordering (one that is close to Pareto optimal), a 'random' curve that orders the operations randomly, and an 'adversarial' curve that represent an ordering resulting in a poor trade-off. The term 'quality' is left general on purpose, as there are many ways to measure quality depending on context.

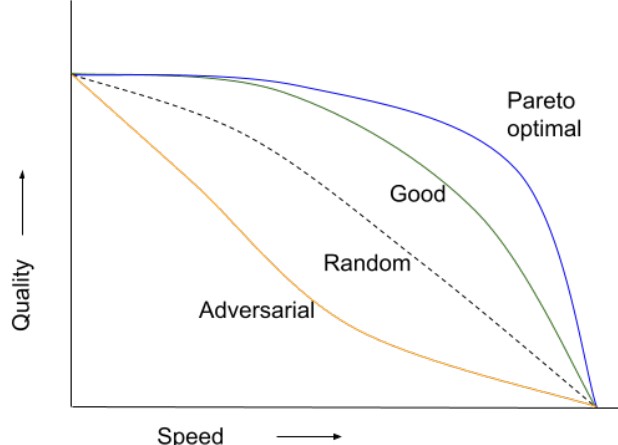

### F. Refinements

Although many methods are known to improve quantization quality, they may incur an additional expense. For this reason, a profile-driven methodology can be useful for determining an appropriate tradeoff.

PGQ allows for incorporating further refinements to boost the default quantization capabilities. In particular, we can incorporate several known quantization techniques by specifying them at the operation level, including:

- Smoothquant [34].
- Subchannel, sometimes known as blocks or groups [11].
- Clipping [1], [21], [36].
- Asymmetric [15], [33] (zero point, or affine-transformation) quantization.

PGQ can be used to generate recipes that specify using different techniques/formats, for example by only applying a certain method with the SQNR is too low, instead of simply turning off quantization as we have shown in the example in this paper.

### G. Compiler Integration

Our compiler integration sits at the graph level provided by modern machine learning compilers such as TVM [8], XLA [20], or ONNX [3]. There are two phases to apply PGQ: (i) instrumentation of tensor statistics data and tensor metadata, and (ii) graph rewriting for applying quantization. From the user's point of view, either stage can be done simply by passing command-line flags when launching the

TABLE I: **Performance gains in training.** Recipe $F$ quantizes compute-bound forward operations only using round-to-nearest (RN). $F+B$ quantizes both forward and backward operations using RN. $F+B_{SR}$ Quantizes both forward (RN) and backward (stochastic rounding). $F+B_{SR}(>1.0)$ and $F+B_{SR}(>0.4)$ are similar to $F+B_{SR}$, but additionally impose a minimum threshold on the SQNR for the backwards pass. $F+B_{SR}(<1.0)$ and $F+B_{SR}(<1.0)$ are the 'adversarial' versions, i.e., imposing a maximum threshold on the operation SQNR.

| Recipe | GemmaV2 2B | | GemmaV2 7B | |
| --- | --- | --- | --- | --- |
| | Step (ms) | Speedup | Step (ms) | Speedup |
| Base | 923.6 | – | 1557 | – |
| $F$ | 905.4 | 2.0% | 1484 | 4.9% |
| $F+B$ | 742.7 | 24.4% | 1317 | 18.2% |
| $F+B_{SR}$ | 745.4 | 23.9% | 1323 | 17.7% |
| $F+B_{SR}(>1.0)$ | 781.8 | 18.1% | 1375 | 13.2% |
| $F+B_{SR}(>0.4)$ | 780.7 | 18.3% | 1372 | 13.5% |
| $F+B_{SR}(<1.0)$ | 867.7 | 6.4% | 1434 | 8.6% |
| $F+B_{SR}(<0.4)$ | 870.4 | 6.1% | 1437 | 8.4% |

model, thereby making it versatile and accessible for users with different levels of expertise.

During the instrumentation stage PGQ inserts specific operations that reduce numerical tensor values into summary statistics and write them to permanent storage as the model is running. While there is additional overhead required for logging tensor statistics, we have found in practice few measurements are usually sufficient. Tensor statistics can include SQNR directly for a specific quantization format, or other quantities from which QNSR can be approximated.

After a quantization recipe is generated offline based on the metadata and summary statistics, we rewrite the network graph with an additional intermediate step (e.g., XLA pass) during the compilation process to change specified operations to lower precision (early on during optimization process to still leverage lower lever compiler optimizations). The graph rewrite involves replacing the full-precision GEMM by: Computing scaling factors along specific dimensions (when quantization is done dynamically), multiplying the inputs by the appropriate scaling factors, casting the inputs into the appropriate formats, performing the lower-precision GEMM, and finally applying the inverse scaling factors on the output. Optionally the other refinements discussed in Section II-F can also be applied at this stage.

## III. PGQ EVALUATION

We apply PGQ to GemmaV2 2B and 7B [28] for a fine-tuning task. The experiments ran on a 64 chip accelerator system using 8x8 topology of V5e TPUs. We did not make any optimizations to the GemmaV2 code, except ensuring the activations were set to bfloat16 before quantization instead of float32. Several recipes are shown and described in Figure 4. We used adversarial recipes for validation as described in Section II-D.

In step 2 of PGQ, quantization is done only for compute-bound operations according to the roofline model to ensure latency benefits, using the recipes we specified. This is done automatically using the metadata collected for each operation.

The (non-adversarial) recipes are presented in the order a user may try: First try the baseline along with several basic non-SQNR strategies (forward only, forward + backward, and forward + backward with stochastic rounding). Comparing these shows most of the performance gains in this particular case come from applying quantization for the backward pass. The last two cases also uncover the cost of stochastic rounding. Since the forward-only recipe did not show degradation while the forward + backward recipes did, the next step would be applying the SQNR strategy on the backwards pass only.

Step 2 of PGQ generates a table with SQNR values that can be inspected by the user. A single operation in the backwards pass had especially low SQNR, which motivates the selection of the 0.4 threshold (the table indicates this was an activation gradient for an operation with a 'get_logits' label). We included the 1.0 threshold as another example. When the operation with SQNR below 0.4 was not quantized, the same quality as baseline was recovered (the same result was found for 1.0). Meanwhile the adversarial recipe *only* quantizing the low-SQNR operation in the backwards pass resulted in poorer quality, confirming it was the primary source of quality loss.

While in the example here we focused on a few concrete threshold values to generate recipes, a more general strategy involves (1) using reasonable priors as a starting point and (2) bisecting to find the most appropriate threshold values for a given use-case. We suggest applying separate thresholds for the forward and backward passes (and applying stochastic rounding for gradients if the format is not wide, resulting in most values underflowing to zero). Typically we found around a 1:1 SQNR ratio is often sufficient for gradients in this setting, while for the forward pass somewhere around 20:1 is often necessary (around 3 on $\ln(1+x)$ scale).

For the SQNR-based thresholds, we selected two threshold values, and also evaluated their adversarial versions (i.e., reverse the decision of whether an operation is quantized or not, in this case restricted to compute-bound gradients). In Figure 4 we show the loss curves of these recipes, and in Table I we show the corresponding step times. Quantizing operations with a SQNR above a low value resulted in curves similar to baseline, while instead quantizing only below a low threshold resulted in divergence similar and instability. Finally, in Figure 5 we combine the performance and quality data for each recipe used for the GemmaV2 7B model, to demonstrate the curves discussed in Section II-D (Pareto optimal vs. adversarial - note that axis are swapped and both axis are reversed, e.g. lower average loss is equivalent to higher quality).

There were only two operations for this particular model with a low SQNR (for both the 2B and 7B models): the embedding table in the forward direction, and the activation gradient from the logits in the backward direction (with values around 0.06 and 0.08, respectively). The forward-direction operation did not impact quality, however the backward direction one clearly did, and seems responsible for all the degradation observed in the recipe quantizing all compute-bound operations in the backwards direction – the adversarial

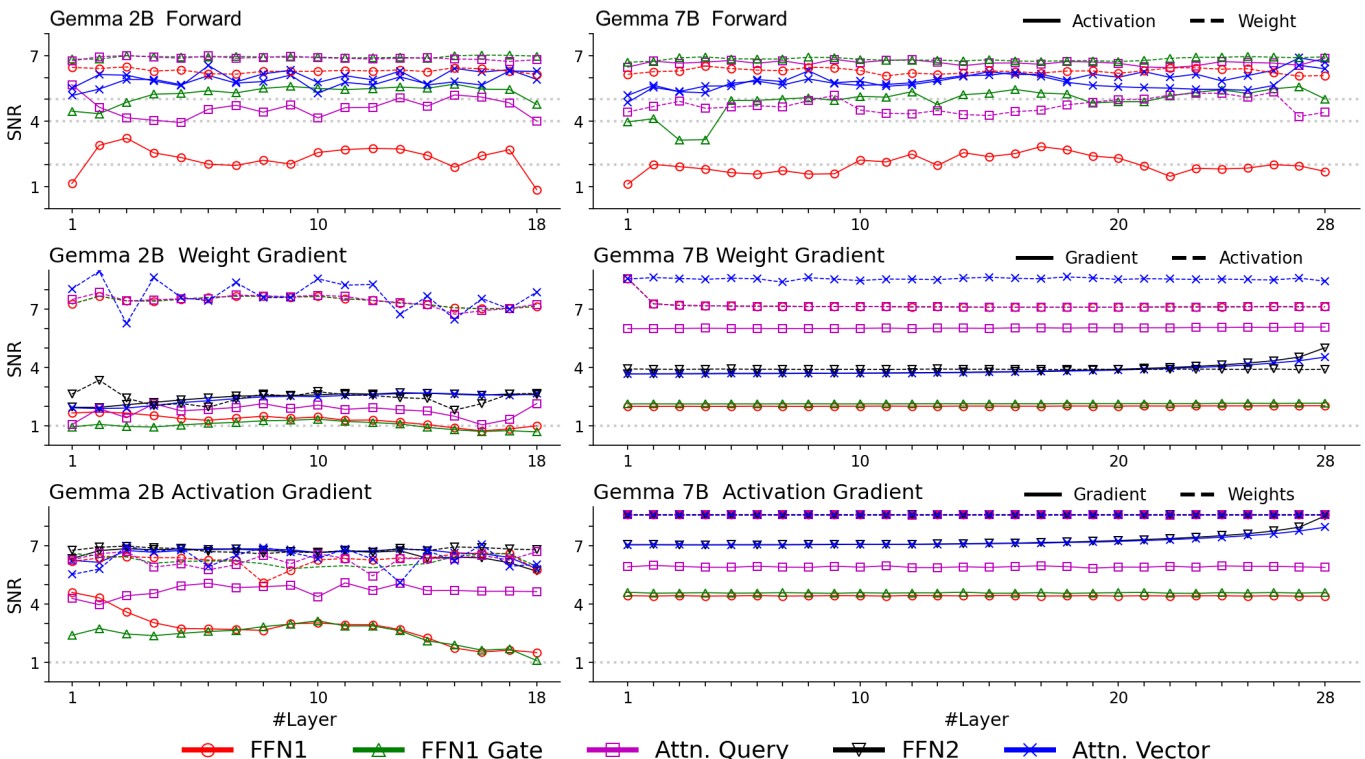

Fig. 3: **Overview of signal-to-quantization-noise** In each subplot we show approximate SQNR on a $\ln(1+x)$ scale for a different set of operations. The first row shows SQNRs collected from the forward pass, while the second and third are gradients. The left and right columns correspond to the 2B and 7B GemmaV2 experiments, respectively. Operations with a higher SQNR tend to result in less quality degradation than those with lower SQNR. For each operation, there are two inputs, shown in the same color/marker style, and distinguished by solid or dashed lines. Our strategy selects a threshold based on the minimum SQNR of the two inputs for a given operation (dotted horizontal lines show a few examples). Weights tend to have the highest SQNR, making them relatively easier to quantize. However, for some operations the SQNR of the inputs may vary significantly between the 2B and 7B versions, and even different layers, with no obvious pattern.

recipes, while obtaining significantly smaller gains in performance, performed as poorly as quantizing all the compute-bound operations.

## IV. CONCLUSIONS

We present PGQ, a user-friendly and extendable method for creating quantization recipes. PGQ is integrated directly into the compiler, thereby making PGQ model and framework agnostic. We demonstrate PGQ's effectiveness on fine-tuning of GemmaV2 2B and 7B models showcasing how PGQ may be used on models representative of real-world workloads.

Some future avenues to enhance the PGQ metric include enriching the sensitivity metric, for example by accounting for inter-layer interaction. Additionally, generating recipes making the best use of various quantization techniques (as discussed in Section II-F) remains non-obvious, for example considering the trade-off between potentially improved quality and extra overhead for using specific techniques.

In conclusion, PGQ solves practical quantization challenges, which arise while applying quantization at scale, such as requirements for extensive user knowledge and black-box uncertainty about quantization decisions. PGQ offers a versatile and accessible framework that empowers users to harness the power of emerging reduced-precision accelerators with minimal initial investment.

## V. ACKNOWLEDGMENTS

The Gemma team at Google who helped us set up open-source models to use as examples in this paper. Naveen Kumar at Google for leadership and support for the project.

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
