# OpenReview forum: "Profile-Guided Quantization: a compiler solution to automate quantization for efficient LLM training"
_iscaconf.org/ISCA/2025/Workshop/MLArchSys — MLArchSys 2025 Oral_

### Official Review · Reviewer_rtAa · 2025-05-12
**Novelty and evaluation result are not clear**

**Confidence:** 2
**Rating:** 4

**Detailed Feedback And Questions For Authors:**

Although this paper gives a strong survey on prior work and a detailed numerical analysis (e.g., Fig. 2) to motivate its importance, I had three concerns described below.

**Novelty of the work is not directly stated**:
This paper can be made stronger by claiming its novelty against the prior work in a more direct manner.
In particular, some "approaches have leveraged sensitivity metrics" as the authors point out in Introduction.
This makes me wonder why these approaches are not sufficient and how this paper (which uses SQNR as the sensitivity metric) is novel against them. The sentence starting from "Noticeably, ..." is vague as it lists up many things, and does not convince exactly why the existing methods do not work.

**Evaluation result is unclear**:
The reasons behind some specific numbers in the evaluation are quite unclear.
How did you do "examining the SQNR values" and found "a single operation in the backwards pass had especially low SQNR"?
I guess the only source of this kind of information is Fig. 2, but it is not mentioned in the evaluation section.
This confusion makes the evaluation quite unclear because now I cannot see how the recipes in TABLE I are determined given the unexplained threshold numbers (e.g., 1.0).

**Compiler integration solely relies on effort of other people**:
One novelty of this work is its compiler integration as it is also prominent in the paper title.
To this end, the paper should explain more on the challenges and the solutions in integrating the method to the compiler.
What I understood from the current paper is that the integration was easy because the existing compilers already supported graph-level instrumentation and modification.
However, these are made possible by the effort of other people and not the novelty of this work.

**Top Reasons To Accept The Paper:**

- Detailed numerical analysis
- Strong survey on prior work

**Top Reasons To Reject The Paper:**

- Novelty of the work is not directly stated
- Evaluation result is unclear
- Compiler integration solely relies on effort of other people

---

### Official Review · Reviewer_EQfD · 2025-05-12
**Promising compiler-based quantization framework**

**Confidence:** 4
**Rating:** 5

**Detailed Feedback And Questions For Authors:**

**Strengths**

- The integration into ML compilers (e.g., TVM/XLA/ONNX) is highly practical and aligns with industry deployment practices.

- The paper leverages SQNR as a local, interpretable metric to automate per-operation quantization decisions, reducing trial-and-error tuning.

- Experiments on GemmaV2 2B and 7B show clear improvements in training speed (up to ~18%) with minimal degradation when SQNR filtering is applied, especially in the backward pass.

- The use of adversarial recipe comparison is a thoughtful addition for empirically validating the effectiveness of the chosen metric.

**Weaknesses**
- Missing clarity on the quantization bit-width used in the experiments (e.g., 4-bit, 8-bit), which is essential for interpreting SQNR effects and reproducibility.

- Limited evaluation scope: the method is only tested on a single model family (GemmaV2), and there is no comparison to other quantization techniques like SmoothQuant or GPTQ/PTQ+QAT toolchains.

- While PGQ is claimed to be framework-agnostic, there is no discussion of integration overhead or support across differing model architectures (e.g., convolutional or MoE-based).

**Suggestions for Improvement**

- Clearly specify the bit-width and quantization format used in each experiment (e.g., int4, int8, FP8).

- Provide a comparison against baseline post-training quantization (PTQ) methods or existing compiler-aware quantization toolchains.

- Extend experiments to more model architectures or training scenarios (e.g., vision transformers, MoE models, or language pretraining from scratch).

**Top Reasons To Accept The Paper:**

- The paper presents a practical and extensible compiler-level quantization framework that minimizes the need for user intervention, lowering barriers to deploying quantized training.

- The use of SQNR-based local metrics provides a compelling method to automate quantization decisions and is supported by adversarial evaluations.

- The method demonstrates up to 18.2% speedup on real-world fine-tuning workloads (GemmaV2 models) without significant quality degradation, indicating practical viability.

**Top Reasons To Reject The Paper:**

- The paper does not clearly specify the bit-width or quantization format used in the experiments, which makes interpretation and reproduction of the results difficult.

- While speedup is demonstrated (up to 18.2%), there is no evaluation of model quality on standard academic benchmarks (e.g., MMLU, ARC, or GLUE) to quantify the accuracy trade-offs, making it unclear whether the speed gains come at the cost of generalization or robustness.

---

### Official Review · Reviewer_1SU6 · 2025-05-16
**A Profile-Guided Quantization Approach with Limited Generalization Evidence**

**Confidence:** 2
**Rating:** 6

**Detailed Feedback And Questions For Authors:**

1. The process of selecting the local metric threshold lacks clarity and generality. Providing a more systematic or automated threshold selection process would strengthen the method
2. Adding a workflow diagram could significantly help readers understand how the proposed profile-guided quantization is applied in practice.
3. While the quantization recipe is applied offline, it would be useful to analyze the overhead of enabling profiling and integrating the quantization recipe into the compilation process.
4. The discussion on how different threshold values (e.g., <0.4 vs. <1.0) impact the final performance is weak and unconvincing. A deeper analysis, theoretical explanation, or more explorations on different threshold values would make the impact of threshold on performance speedup more pronounced.

**Top Reasons To Accept The Paper:**

1. Proposed an effort-less quantization method at compile time, reducing the need for runtime intervention or manual tuning
2. Most parts of the paper are easy to follow, except the experiment section
3.The introduced a local metric for guiding automatic quantization, could be a potentially reusable idea for other optimization tasks beyond quantization.

**Top Reasons To Reject The Paper:**

1. Not showing how general this method can be applied to other models, considering only providing insights based on 1 model architecture. Another representative model should be considered
2. When choosing the threshold for the local metric, it is like a more empirically based solution, which may require intensive experiments. This might be contradicting from what authors proposed.
3. Experimental results, especially Figure 3, are hard to interpret and not self-explanatory, making it difficult to evaluate the effectiveness of the method.

---

### Official Review · Reviewer_CcSM · 2025-05-18
**This paper introduces Profile-Guided Quantization (PGQ), a compiler-level system that automatically determines where and how to apply mixed-precision quantization in large language model training by profiling per-operator signal-to-quantization-noise ratios (SQNRs). Without modifying model code or framework APIs, PGQ instruments the computation graph to collect activation and gradient statistics, generates a “quantization recipe” based on SQNR thresholds, and rewrites the graph to insert scale factors, casts, and low-precision kernels. Evaluated on GemmaV2 2B and 7B models, PGQ achieves up to 24.4% (2B) and 18.2% (7B) step-time speedups during fine-tuning with negligible impact on model accuracy.**

**Confidence:** 3
**Rating:** 6

**Detailed Feedback And Questions For Authors:**

1, Could you include GPU and CPU benchmarks (e.g., NVIDIA A100, AMD MI200) to validate that the PGQ pipeline yields similar speedups beyond TPU?

2, Have you considered integrating a small held-out calibration pass or lightweight hyperparameter search to automatically pick the optimal SQNR threshold?

3, Please report the extra wall-clock time, memory, and storage cost of running the preguided profiling phase, and discuss any strategies to hide or parallelize that overhead.

4, How might PGQ be adapted to work with dynamic frameworks (e.g., PyTorch eager mode or TensorFlow’s TF-Function), where graph construction and execution interleave?

**Top Reasons To Accept The Paper:**

Strengths:

1, Combines local SQNR profiling with compiler graph-rewriting to automate mixed-precision decisions, reducing manual tuning effort and search space.

2, Framework- and Model-Agnostic：Operates purely at the graph/IR level—users only toggle a compiler flag, with no changes required to model definitions or training scripts.

3, Demonstrates 17–24% speedup on real LLM fine-tuning tasks.

**Top Reasons To Reject The Paper:**

Weaknesses:

1, Experiments run solely on TPU V5e clusters; no GPU, CPU, or other accelerator results to confirm generality.

2, SQNR cutoffs still require manual selection or light tuning per model/task; no automated threshold-search strategy is proposed.

3, The paper does not quantify or amortize the runtime and memory overhead introduced by profiling inserts.

4, The approach is demonstrated on static graphs; it remains unclear how PGQ would integrate with PyTorch’s eager or hybrid execution modes.